# Effect of NO_x_ and NO_2_ Concentration Increase in Ambient Air to Daily Bronchitis and Asthma Exacerbation, Silesian Voivodeship in Poland

**DOI:** 10.3390/ijerph17030754

**Published:** 2020-01-24

**Authors:** Małgorzata Kowalska, Michał Skrzypek, Michał Kowalski, Josef Cyrys

**Affiliations:** 1Department of Epidemiology, School of Medicine in Katowice, Medical University of Silesia, 40-055 Katowice, Poland; mkowalska@sum.edu.pl; 2Department of Biostatistics, School of Health Sciences in Bytom, Medical University of Silesia, 40-055 Katowice, Poland; mskrzypek@sum.edu.pl; 3Environmental Exposure Assessment Group, Institute of Epidemiology, Helmholtz Zentrum München, 85764 Neuherberg, Germany; cyrys@helmholtz-muenchen.de

**Keywords:** nitrogen oxides, particulate matter, acute respiratory effect, time series study

## Abstract

There is a discussion in Europe about the dominant role of air pollution for health effects, most researchers claim that the particulate matter is responsible for inflammatory processes in the respiratory system, while others underline the role of nitrogen dioxide. The aim of the study was to assess the risk related to NO_x_, NO_2_ and PM_2.5_ concentration increase and daily outpatient visits or hospitalization due to bronchitis and asthma exacerbation in the entire population of Silesian Voivodeship, Poland. To assess the relationship between daily pollutants concentrations and the number of outpatient visits or hospitalizations due to bronchitis and asthma (available in the regional registry), the multivariable log-linear Poisson regression model was used. Results were presented by relative risk (RR) of health outcomes related to the increase in pollutant concentration by unit (interquartile range). Obtained results confirmed a statistically significant association between outpatient visits and hospitalizations due to bronchitis and asthma exacerbation and daily nitrogen oxides concentrations in Silesian voivodeship, Poland. The strongest relationship was observed in the case of NO_2_ and outpatient visits due to bronchitis, e.g., RR = 1.434 (1.308–1.571) for exposure expressed by the 50-day moving average concentration. In the case of hospitalizations, the health effect was lagged a few days in relation to the increase in exposure.

## 1. Introduction

The quality of ambient air expressed by particulate matter (PM_10_ and PM_2.5,_ coarse and fine, respectively) and nitrogen dioxide (NO_2_) concentration in Europe is steadily improving, however current reports suggest that Poland is one of the countries where the level of pollutants are exceeding the EU limit values for PM_10_ and PM_2.5_ as well as NO_2_ introduced for protecting human health [1]. While in the case of fine dust exposure (PM_2.5_) the number of premature deaths in Poland is one of the highest in Europe, in the case of exposure to NO_2_ the health effects in Poland are definitely lower [1]. The unfavourable situation was recognized in southern part of the country, such as Lesser Poland (Małopolska) and Silesian Voivodships for many reasons [2]; first because of the unfavourable geographical location (lowland surrounded by hills), secondly because of the high density of population living in both regions [3] and also high traffic intensity [4,5], finally because of pronounced coal combustion in individual heating stoves in this area [1,6]. 

The main task of the current study was to assess the risk related to NO_x_ and NO_2_ concentration increase and daily outpatient visits or hospitalization due to bronchitis and asthma exacerbation in the entire population of Silesian Voivodeship. Results of the previous study confirmed, that daily outpatient visits and hospitalizations due to respiratory diseases significantly increase with the worsening of ambient air quality due to an increase of PM_10_ and PM_2.5_ concentration during the cold season in the study area [2]. Among all NO_x_ and NO_2_ sources motor vehicle emission and also energy production are the most important [1]. Consequently, in the group of potential short-term effects of exposure to NO_x_ and NO_2_ are following outcomes: inflammatory reactions of the lungs and bronchi, respiratory symptoms such as coughing, wheezing, and problems with breathing, especially in patients with earlier recognized asthma or other chronic respiratory diseases [7]. Recently, there is a discussion in Europe about the dominant role of air pollution for health effects, most researchers claim that the particulate matter is responsible for inflammatory processes in the respiratory system, while others underline the role of nitrogen dioxide [8]. Results of previous studies suggest clear evidence of PM_10_ effects on the occurrence of asthma symptom episodes, however in the case of NO_2_ exposure it is difficult to interpret obtained results due to health effects depending on the examined time lags [7,9]. Moreover, in our study, the estimated Spearman correlation coefficients between NO_x_, NO_2_ and particulate matter concentration in ambient air were very high, which makes the separation of the health effects for different pollutants difficult [2].

We believe that the results of our study will help to assess the separate risk of daily outpatient visits and hospitalization due to bronchitis and asthma exacerbation in relation to nitrogen oxides exposure. We understand that obtained results can help to improve the inhabitants’ knowledge on the real hazard, and will reinforce the social activity needed to improve the quality of ambient air, including acceptance for the need of modernization the heating system and the restriction of individual road transport in crowded cities. 

## 2. Materials and Methods

Data on daily registered outpatient visits or hospitalizations due to bronchitis (J20-J21 by ICD-10) and asthma exacerbation (J45–J46 by ICD-10) in the study period (1 January 2016 to 31 August 2017) were obtained from the National Health Fund database in Katowice. The final database included health outcomes registered for about 2.5 million inhabitants of the central agglomeration area located in the Silesian voivodeship [2]. The total number of outpatient visits due to bronchitis and asthma exacerbation in the study period was the highest in the winter period and the lowest in the summertime. Table 1 shows detailed, seasonal variability of respiratory outcomes including, outpatient visits and hospitalization.

Air pollution and meteorological data in the study period, including a daily concentration of NO_2_, NO_x_, PM_2.5_, PM_10_ in ambient air, as well as daily temperature, relative humidity, and atmospheric pressure, were gathered from the automatic measurement stations of Provincial Inspectorate of Environmental Protection in Katowice database [10]. Detailed meteorological situation in the studied region and daily average concentrations of pollutants in the chosen period was described in the previous publication [2]. It is worth mentioning that the correlations between NO_x_ and other ambient air pollutants were statistically significant and stronger in winter than in the summer period (Table 2).

To assess the relationship between NO_x_, NO_2_, PM_2.5_ and the number of outpatient visits or hospitalizations due to bronchitis and asthma, the multivariable log-linear Poisson regression model was used. The model was linked with the following equation:
log[E(ND)] = Xβ(1)
where E(ND) is a dependent value (observed a daily number of outpatient visits or hospitalizations), X is a vector of independent variables (daily concentration of NO_x_, NO_2_ or PM_2.5_), and β means the calculated regression coefficient. The variables describing ambient air pollutants were expressed as 1, 3, 5, 7, 14, 30, 40, and 50-day moving averages of NO_x_, NO_2_ and PM_2.5_ concentrations. Moreover, a group of independent variables includes confounding factors such as daily average ambient air temperature, atmospheric pressure, relative humidity, season of the year, influenza episodes, and additionally weekend days (working day vs. holiday) for outpatient visits. Results of the multivariable analysis were presented by relative risk (RR) of bronchitis or asthma exacerbation related to the increase in NO_x_, NO_2_ and PM_2.5_ concentration by unit (interquartile range—IQR_NOx_ = 26.17 µg/m^3^ or IQR_NO2_ = 12.67 µg/m^3^ and IQR_PM2.5_ = 22.5 µg/m^3^, respectively). RR was calculated using the following equation:RR = exp (β × IQR)(2)
where β is the expected regression coefficient. The assumed level of statistical significance used was α = 0.05, calculations were conducted using SAS version 9.4 (SAS Institute Inc., Cary, NC, USA).

## 3. Results

Obtained results revealed the worse quality of ambient air at the beginning of the 2017 year, with the highest NO_x_ and NO_2_ concentration in the period from the 07 to 11 January (Figure 1). Similarly, daily PM_2.5_ concentration was the highest in the same period; detailed data on air quality were presented in a previous publication [2]. Obtained data confirmed the occurrence of two winter smog episodes in the study period, both of them were in January (4–7 January 2016 and 7–11 January 2017) in which the average temperature of ambient air was the lowest. Moreover, the median (and IQR) value of pollutants concentrations were the highest in the winter season (from 21 December to 19 March), with the following values: 44 (49.0) µg/m^3^ for PM_2.5_ and 52.58 (53.22) µg/m^3^ for PM_10_, 41 (46.9) µg/m^3^ for NO_x_ and 28.7 (16.9) µg/m^3^, respectively.

Table 3 and Figure 2, Figure 3 and Figure 4 illustrate the results of multivariable analysis assessing the risk ratio of daily outpatient visits or hospitalization due to bronchitis and asthma exacerbation related to the pollutants’ concentration increase by the unit (interquartile range value—IQR). Exposure was expressed by the moving average concentration. It is interesting that the relationship between nitrogen oxides and bronchitis risk ratio was statistically significant in every time lag, assumed way of exposure assessment (including the 1-day moving average concentration). In the case of PM_2.5_ a significant effect was observed only for longer exposure periods (starting from the 5-day moving average concentration in case of outpatient visits and 14-days moving average concentration for hospitalization). Similarly, in the case of asthma exacerbation statistically significant dose-response was observed for all NO_x_ and NO_2_ averaged exposure period, and for both health outcomes (outpatient visits and hospitalization). The relationship between PM_2.5_ exposure and respiratory health problems was weaker and statistically significant only for longer exposure periods.

## 4. Discussion

The results obtained in our study confirmed that elevated NO_x_ or PM_2.5_ ambient air concentrations caused an increase of daily outpatient visits or hospitalization due to bronchitis or asthma exacerbations. Despite the significant correlation between the concentrations of both pollutants (Spearman correlation coefficient *r* = 0.76 in the total study period, and *r* = 0.86 in wintertime), it was found that nitrogen oxides caused an earlier and stronger health response to an increase in average moving exposure per unit (expressed by IQR). A similar observation was provided by a study conducted in Vietnam, where the strongest effect was observed for NO_2_ and hospital admissions due to pneumonia in children under 18 years of age, obtained risk ratio was RR = 1.061 (95% CI: 1.025–1.098) [11]. Simultaneously, a statistically significant effect was observed for bronchitis and asthma hospitalizations in children aged 0–17 years, RR = 1.055 (95% CI: 1.004–1.108) for NO_2_ and RR = 1.056 (95% CI: 1.004–1.111). It is worth to note, that nitrogen oxide was a dominant pollutant in ambient air, related to the development of asthma in children in the southern part of Sweden, a region with a significantly better quality of air than in our study area [12]. Moreover, in our study, the statistically significant relative risk for NO_2_ and NO_x_ already occurs during one-day moving average concentration and is gradually increasing with a longer time of exposure. Such observation confirmed earlier results obtained for PM_2.5_—each increase of dose, expressed as moving average concentration over a long time, leads to an increase in a daily number of outpatient visits or hospitalization due to acute respiratory diseases [2]. Current data suggest that the stronger relationship was observed in the case of outpatient visits due to bronchitis than due to asthma exacerbation. Furthermore, the hospitalization health effect was lagged a few days in relation to the increase in exposure. This observation is consistent with the results of the previously cited work, the time window from exposure until day of pneumonia hospitalization includes even one week [11]. A similar observation was reported in Canada, significant associations between NO_2_ and emergency department visits due to upper respiratory diseases were reported for a time lag of 8 days [13]. 

The relationship between daily levels of nitrogen dioxide and particulate matter in ambient air and hospitalization for acute respiratory infection has been reported also in an earlier publication [14]. Barnett et al. confirmed the largest association of asthma admissions among children (5–14 years) in relation to an increase in daily NO_2_ concentration (6% in each increase of NO_2_ level by 5.1 ppb). The authors suggested that NO_2_ exposure makes people more susceptible to respiratory viral infections that exacerbate asthma, and probably gas is a precursor of photochemical smog leading to the formation of reactive ozone and secondary particles. 

Moreover, it is well known that traffic-related air pollution (such as NO_x_, NO_2_, ultra-fine particles) can trigger asthma symptoms in human beings [15]. Current Swedish publication suggests that the implementation of exhaust-free transport would lead to a reduction in NO_2_ emission and in consequence to a significant decrease (by 10%) of bronchitis in asthmatic children, even in one of the cleanest region of Europe [16]. Similarly, the results of epidemiological studies in Japan revealed that traffic-related air pollution was associated with the persistence of respiratory disorders such as asthmatic symptoms in preschool children. The situation has improved since 2001 because of the Automobile NO_x_/PM law in Japan. Measured nitrogen oxides and particulate matter concentrations have gradually decreased and the prevalence of respiratory and allergic disorders in 3-year-old children was lowering [17]. Our own observations can be used to underline the need for the modification of car traffic in such large agglomerations as the Silesian voivodeship. 

Recent publications indicate that road transport remains the main source of nitrogen oxides emissions. However, the observed pattern of the daily concentration of NO_2_ and NO_x_ in the studied region (Silesian voivodeship) indicates a significant contribution of following emission sources, such as energy production (72.39% of total NO_x_ emission), vehicle transport (22.19%), individual heating systems (4.52%), and others sources (0.89%) [18]. Moreover, the highest level of both gases emission and peak concentration was noted in January 2017, i.e., the period related to winter smog in Poland [2], which confirms the importance of heating production. Promising is an activity of local authorities in one of the largest city located in the studied region (Katowice) associated with the implementation of new, more effective indicators of ambient air quality required to better protection of citizen health [19]. We believe that a more aware community will be ready to accept both, the need to modernize the heating system, including reducing coal combustion in individual heating, as well as improvements of the public transport system, which would lead to increased use of this system.

The presented study has some limitations. First, an ecological type of study (time-series analysis) relies on pollution data (PM_2.5_, NO_x_, and NO_2_) obtained from stationary monitoring stations giving no possibility to control personal exposure. However, the lack of other measurement data and the lack of evidence from cohort studies in Poland justify the chosen way of assessment, moreover, this method of research is widely used in environmental epidemiology [20]. Secondly, the available registry of health outcomes relies on the diagnosis available in ICD-10 coding and we are aware of some mistakes or difficulties in diagnosing diseases such as bronchitis, especially in the youngest patients or in those with overlapping respiratory diseases. Next, the daily number of hospital admission due to asthma was rather small in Poland probably because the patients have good treatment and they control their health status. The number of daily bronchitis visits and hospitalizations was higher and the obtained relative risk was more reliable than in the case of asthma exacerbation. Finally, we can’t forget that usually environmental stressors show effects delayed in time, and dose–response analysis require proper statistical model. Our analysis was based on the multivariable log-linear Poisson regression model, which seems too simple for some epidemiologists. Promising proposition for the future analyses could be a new more flexible variants of distributed lag non-linear models (DLNM) implemented in the statistical environment R [21]. The more, so that the observation period is quite long and there was a visible deterioration in the quality of ambient air between 7 and 11 January 2017 we decided to reassess the RR value for two separate periods (from January 1 to August 30 of each year), respectively. Obtained results show some differences in the picture of dose-response for each year (detailed results are available upon request). The relationship between outpatient visits due to bronchitis and asthma exacerbation and NO_2_ or NO_x_ concentration was similar in 2017 and in whole studied period. Additionally, a statistically significant effect of fine particle on outpatient visits was revealed, including all moving average concentration of pollution. On the second hand, the lower concentrations in 2016 resulted in significant reduction of RR value. In the case of hospitalization, values of RR were not statistically significant in longer time exposure than this, expressed by 1 or 3 day-moving average concentration. In conclusion, a shorter time period results in fewer recorded health events, which can lead to changes in RR values.

## 5. Conclusions

Summarizing, the results of our study have shown a statistically significant association between outpatient visits and hospitalizations due to bronchitis and asthma exacerbation and daily nitrogen oxides concentrations in Silesian voivodeship, Poland. The strongest relationship was observed in the case of NO_2_ and outpatient visits due to bronchitis. In the case of hospitalizations, the health effect was lagged a few days in relation to the increase in exposure. 

## Figures and Tables

**Figure 1 ijerph-17-00754-f001:**
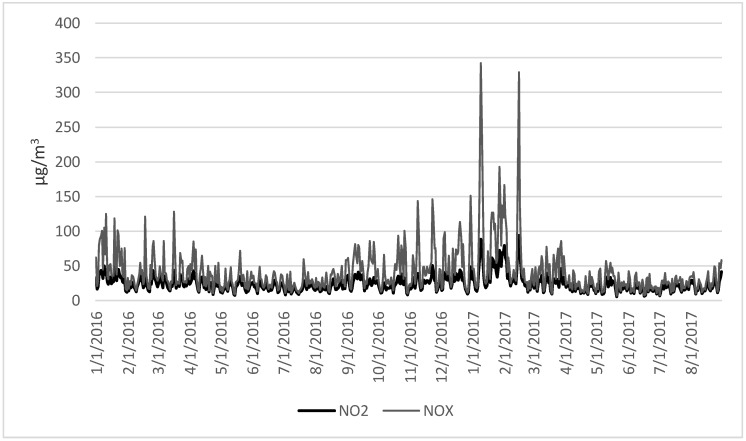
Daily concentration of NO_x_ and NO_2_ in ambient air in the period 1 January 2016 to 31 August 2017, Silesian voivodeship.

**Figure 2 ijerph-17-00754-f002:**
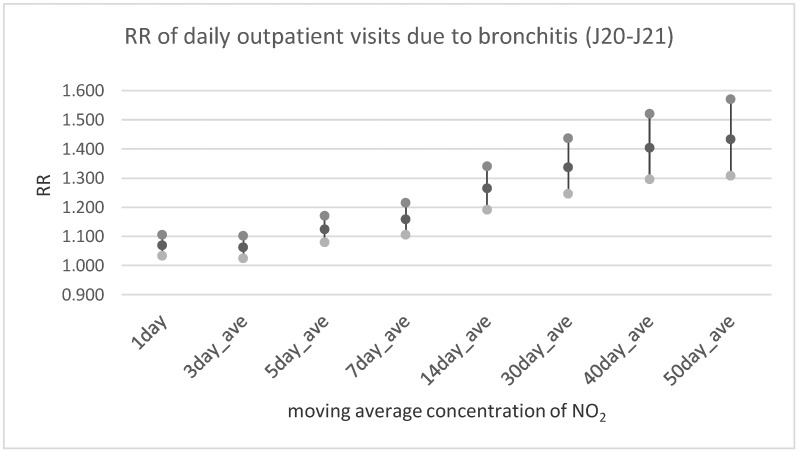
The relative risk of daily outpatient visits due to bronchitis (J20–J21 according to ICD-10) in relation to an increase of NO_2_ concentration by IQR = 12.67 µg/m^3^.

**Figure 3 ijerph-17-00754-f003:**
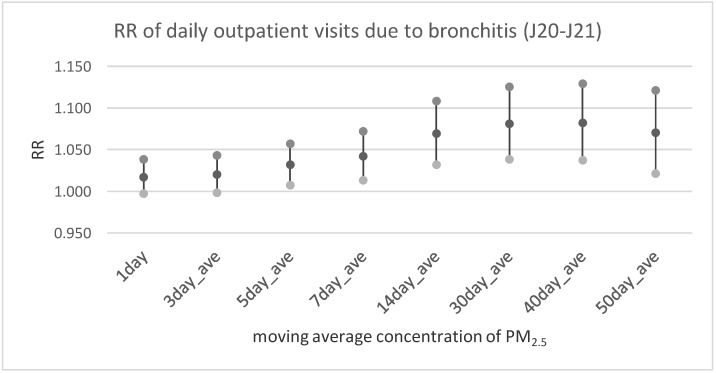
The relative risk of daily outpatient visits due to bronchitis (J20–J21 according to ICD-10) in relation to an increase of PM_2.5_ concentration by IQR = 22.5 µg/m^3^.

**Figure 4 ijerph-17-00754-f004:**
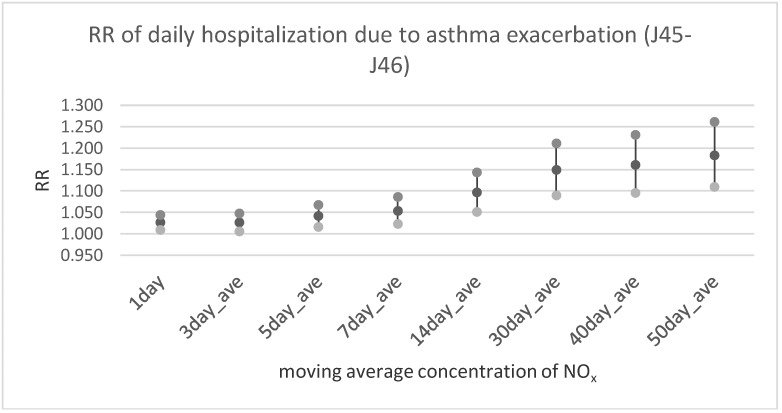
The relative risk of hospitalizations due to asthma exacerbation (J45–J46 according to ICD-10) in relation to an increase of NO_x_ concentration by IQR = 26.17 µg/m^3^.

**Table 1 ijerph-17-00754-t001:** The number of respiratory outcomes in the central agglomeration of Silesian voivodeship registered from the 1 January 2016 to the 31 August 2017, by a season of the year.

	Total Number of Respiratory Outcomes in a Particular Season of the Study Period (1 January 2016 to 31 August 2017)				
	Total	Winter 21 Dec–21 Mar	Spring 22 Mar–20 Jun	Summer 21 Jun–21 Sep	Autumn 22 Sep–22 Dec
**Outpatient visits total J00–J99**	3,550,901	1,506,255	929,322	327,530	787,794
**Outpatients visits bronchitis J20–J21**	336,106	173,781	72,921	20,805	68,599
**Outpatient visits asthma J45–J46**	156,836	49,068	49,267	24,331	34,170
**Hospitalization total J00–J99**	60,346	21,576	17,292	9,268	12,210
**Hospitalization bronchitis J20–J21**	4674	2540	1051	292	791
**Hospitalization asthma J45–J46**	3815	1209	1168	635	803

**Table 2 ijerph-17-00754-t002:** Correlation coefficients for compared determinants of aerosanitary situation in the study period (1 January 2016 to 31 August 2017), Silesian voivodeship.

Compared Parameters	Total Period (1 January 2016 to 31 August 2017)	Winter	Summer
**NO_x_ vs. NO_2_**	0.96 (*p* < 0.001)	0.97 (*p* < 0.0001)	0.94 (*p* < 0.0001)
**NO_x_ vs. O_3_**	−0.59 (*p* < 0.001)	−0.72 (*p* < 0.0001)	−0.04 (*p* = 0.6)
**NO_x_ vs. PM_10_**	0.82 (*p* < 0.001)	0.91 (*p* < 0.0001)	0.70 (*p* < 0.0001)
**NO_x_ vs. PM_2.5_**	0.76 (*p* < 0.001)	0.86 (*p* < 0.0001)	0.69 (*p* < 0.0001)
**NO_x_ vs. SO_2_**	0.61 (*p* < 0.0001)	0.82 (*p* < 0.0001)	0.33 (*p* = 0.0001)
**NO_x_ vs. air temperature**	−0.34 (*p* < 0.0001)	−0.46 (*p* < 0.0001)	0.36 (*p* < 0.0001)
**NO_x_ vs. relative humidity**	0.13 (*p* = 0.001)	0.12 (*p* = 0.1)	−0.22 (*p* = 0.01)

**Table 3 ijerph-17-00754-t003:** The relative risk of daily outpatient visits or hospitalization due to bronchitis and asthma exacerbation related to the increase of NO_x_, NO_2_ and PM_2.5_ moving average concentration by interquartile range (IQR) value.

Moving Average Concentration [µg/m^3^]	Daily Health Outcome	Relative Risk (RR) and 95% Confidence Interval (CI) Related to an Increase of		
		NO_2_ Concentration by IQR = 12.67 µg/m^3^	NO_x_ Concentration by IQR = 26.17 µg/m^3^	PM_2.5_ Concentration by IQR = 22.5 µg/m^3^
**Bronchitis (J20–J21)**				
**1day**	out. visits	1.070 (1.035–1.106)	1.027 (1.007–1.049)	1.017 (0.997–1.038)
	hospitalization	1.060 (1.016–1.106)	1.015 (0.987–1.044)	0.992 (0.965–1.020)
**3day_ave**	out. visits	1.063 (1.025–1.103)	1.025 (1.000–1.050)	1.020 (0.998–1.043)
	hospitalization	1.072 (1.023–1.124)	1.026 (0.993–1.061)	1.001 (0.972–1.031)
**5day_ave**	out. visits	1.125 (1.080–1.172)	1.063 (1.033–1.095)	1.032 (1.007–1.057)
	hospitalization	1.124 (1.068–1.183)	1.059 (1.020–1.099)	1.017 (0.985–1.051)
**7day_ave**	out. visits	1.160 (1.107–1.215)	1.097 (1.059–1.136)	1.042 (1.013–1.072)
	hospitalization	1.161 (1.096–1.229)	1.088 (1.041–1.136)	1.031 (0.995–1.068)
**14day_ave**	out. visits	1.265 (1.193–1.342)	1.210 (1.152–1.270)	1.069 (1.032–1.108)
	hospitalization	1.269 (1.182–1.363)	1.179 (1.111–1.252)	1.063 (1.018–1.109)
**30day_ave**	out. visits	1.339 (1.247–1.437)	1.293 (1.218–1.373)	1.081 (1.038–1.125)
	hospitalization	1.457 (1.342–1.581)	1.336 (1.246–1.433)	1.132 (1.079–1.187)
**40day_ave**	out. visits	1.405 (1.297–1.522)	1.344 (1.258–1.437)	1.082 (1.037–1.129)
	hospitalization	1.578 (1.438–1.733)	1.447 (1.337–1.565)	1.169 (1.112–1.231)
**50day_ave**	out. visits	1.434 (1.308–1.571)	1.360 (1.262–1.467)	1.070 (1.021–1.121)
	hospitalization	1.669 (1.502–1.855)	1.510 (1.383–1.647)	1.188 (1.125–1.255)
**Asthma exacerbation (J45–J46)**				
**1day**	out. visits	1.051 (1.024–1.080)	1.026 (1.009–1.044)	1.009 (0.992–1.027)
	hospitalization	1.189 (1.125–1.256)	1.086 (1.048–1.126)	1.025 (0.986–1.065)
**3day_ave**	out. visits	1.044 (1.013–1.075)	1.026 (1.005–1.047)	1.011 (0.993–1.031)
	hospitalization	1.089 (1.018–0.164)	1.048 (0.999–1.098)	1.019 (0.975–1.064)
**5day_ave**	out. visits	1.067 (1.030–1.104)	1.041 (1.016–1.068)	1.017 (0.995–1.038)
	hospitalization	1.077 (0.997–1.163)	1.044 (0.986–1.105)	1.026 (0.977–1.076)
**7day_ave**	out. visits	1.078 (1.036–1.122)	1.054 (1.022–1.086)	1.019 (0.994–1.044)
	hospitalization	1.115 (1.023–1.214)	1.071 (1.003–1.144)	1.053 (0.979–1.090)
**14day_ave**	out. visits	1.129 (1.073–1.187)	1.096 (1.051–1.143)	1.031 (1.001–1.063)
	hospitalization	1.175 (1.054–1.310)	1.145 (1.047–1.253)	1.056 (0.989–1.128)
**30day_ave**	out. visits	1.182 (1.111–1.258)	1.149 (1.090–1.211)	1.045 (1.013–1.082)
	hospitalization	1.245 (1.090–1.422)	1.218 (1.090–1.361)	1.061 (0.985–1.143)
**40day_ave**	out. visits	1.204 (1.123–1.290)	1.161 (1.095–1.231)	1.050 (1.012–1.089)
	hospitalization	1.264 (1.086–1.471)	1.225 (1.078–1.391)	1.061 (0.978–1.151)
**50day_ave**	out. visits	1.229 (1.139–1.325)	1.183 (1.110–1.261)	1.056 (1.016–1.098)
	hospitalization	1.365 (1.157–1.611)	1.304 (1.135–1.498)	1.069 (0.981–1.166)

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
