# Peer review of "Effect of NO_x_ and NO_2_ Concentration Increase in Ambient Air to Daily Bronchitis and Asthma Exacerbation, Silesian Voivodeship in Poland"

_ijerph, 2020, doi:10.3390/ijerph17030754_

Round 1
Reviewer 1 Report
The results of our study have shown a statistically significant association between outpatient visits and hospitalizations due to bronchitis and asthma exacerbation and daily nitrogen oxides concentrations in Silesian voivodship, Poland. The strongest relationship was observed in the case of NO2 and outpatient visits due to bronchitis and asthma, Authors propose an explanation suggesting that NO2 exposure makes people more susceptible to respiratory viral infections that exacerbate asthma, and probably gas is a precursor of photochemical smog leading to the formation of reactive ozone and secondary particles. The idea is in agreement with current views on the pathology mechanism of the exacerbations in asthma.
The majorbenefit of the study is the access to daily registered outpatient visits or hospitalizations due to bronchitis (J20-J21 by ICD-10) and asthma exacerbation (J45–J46 by ICD-10) in the study period (1 January 2016 to 31 August 2017) in whole population of Katowice (about 2.5 million inhabitants).
Minor remarks:
First remarks refers to methods of exposure assessment of air pollution and meteorological data in the study period, Data were gathered from the automatic measurement stations of Provincial
Inspectorate of Environmental Protection in Katowice database. Authors report that detailed daily average concentrations of pollutants in the chosen period was described in the previous publication [3]. However, I think the brief description should be provided in this paper as well.
The second remark refers to the source of NO2 and NOx exposure in the studied region. On one hand authors identify as major source a contribution of an energy production (72.39% of total NOx emission), while vehicle transport is a secondary one (22.19%), On other hand they underline the need for the modification of car traffic.
Author Response
Reviewer 1
First remarks refers to methods of exposure assessment of air pollution and meteorological data in the study period, Data were gathered from the automatic measurement stations of Provincial Inspectorate of Environmental Protection in Katowice database. Authors report that detailed daily average concentrations of pollutants in the chosen period was described in the previous publication [3]. However, I think the brief description should be provided in this paper as well.
Thank you for this comment, and we tried to improve the manuscript as expected by adding the following description (line 110-115): ‘Obtained data confirmed the occurrence of two winter smog episodes in the study period, both of them were in January (04-07.01.2016 and 07-11.01.2017) in which the average temperature of ambient air was the lowest. Moreover, the median (and IQR) value of pollutants concentrations were the highest in the winter season (from 21 December to 19 March), with the following values: 44 (49.0) µg/m3 for PM2.5 and 52.58 (53.22) µg/m3 for PM10, 41 (46.9) µg/m3 for NOx and 28.7 (16.9) µg/m3, respectively.’
The second remark refers to the source of NO2 and NOx exposure in the studied region. On one hand authors identify as major source a contribution of an energy production (72.39% of total NOx emission), while vehicle transport is a secondary one (22.19%), On other hand they underline the need for the modification of car traffic.
Thank you for this comment, and we rearranged somewhat the following sentences:
First, in the Introduction (line 65): ‘We understand that obtained results can help to improve the inhabitants’ knowledge on the real hazard, and will reinforce the social activity needed to improve the quality of ambient air, including acceptance for the need of modernization the heating system and the restriction of individual road transport in crowded cities.’
The second, in the Discussion section (line 195-197) :
‘We believe that a more aware community will be ready to accept both, the need to modernize the heating system, including reducing coal combustion in individual heating, as well as improvements of the public transport system, which would lead to increased use of this system.’
Reviewer 2 Report
The authors present a valuable analysis of the health effects of air pollutants on the Silesia Province of Poland. Their findings should be useful to inform risk management approaches. Some additional analysis, explanation, and language editing are required before publication.
Improvements in Section 2. Materials and Methods:
The regression analysis is appropriate but the authors should explain the implications of the high correlations among the pollutants of interest on their analytical approach. Because of the high correlations, they cannot include PM as a independent (or control) variable. This would typically be done in the effort to isolate the effects of a particular pollutant (NOx, NO2). This is discussed by Dominici et al. 2010 (in Epidemiology. 2010 Mar;21(2):187-94. doi: 10.1097/EDE.0b013e3181cc86e8).
The authors make note of the very high air pollution in January 2017. I would like the authors to conduct a sensitivity analysis on the influence of this period on their results overall. They should include results of the sensitivity analysis in the discussion, i.e., does this very high exposure period have a strong influence on their findings, or not?
Presentation issues, language and typos:
Use of subscripts for pollutant abbreviations and units of measurement should be made consistent in text and tables.
Lines 31-32: further describe improvement in air quality. Are conditions in Europe improving on average?
Line 32: Substitute "...is steadily improving" in place of "is systematically improving"
Line 48: "...energy production are the most..."
Line 56: "...health effects depending..."
Line 64: Word choice, suggest "crowded cities" rather than "crowded agglomerations"
Line 69: If the data cover a time period of 2016-2017, why is "(in 2015)" included here? This is not clear.
Line 73: Substitute "including" instead of "occurring both"
Line 76: There should be a break in text after a table, recommend inserting 1-2 blank lines.
Line 79: "Air quality" is more commonly used than "aerosanitary" (also line 107)
Line 86: Insert space after the table.
Line 94: Delete "a" before season
Line 117: Insert "a" before "significant"
Line 120: delete "function"
Line 123: This is Table 3. Suggest presenting results here with two decimal places to improve readability.
Table 3, section on asthma exacerbation, NO2, 3day lag hospitalizations: there is a typo, zero instead of dash
Line 173: Substitute "has improved" in place of "was going better"
Line 184: Insert a period after "production"
Line 191: Use "giving" rather than "gives"
Line 196: Insert "the" before "youngest"
Line 198: "Admissions"
Line 199: "The number of daily bronchitis visits and hospitalizations was higher and the..."
Author Response
Reviewer 2
Improvements in Section 2. Materials and Methods:
The regression analysis is appropriate but the authors should explain the implications of the high correlations among the pollutants of interest on their analytical approach. Because of the high correlations, they cannot include PM as a independent (or control) variable. This would typically be done in the effort to isolate the effects of a particular pollutant (NOx, NO2). This is discussed by Dominici et al. 2010 (in Epidemiology. 2010 Mar;21(2):187-94. doi: 10.1097/EDE.0b013e3181cc86e8).
Thank to the Reviewer for the remark. Once the first version of the manuscript was ready, we performed an additional analysis of the associations between PM2.5 and health outcomes and decided to report the results in the manuscript in addition to the results for NO2 and NOx. However, in the method section we didn’t add PM2.5 to the description (line 90 and 94). This could lead of course to some misunderstandings. We are sorry for that. In truth we didn’t include PM2.5 as an independent variable to the model; each model was calculated separately for only one particular pollutant (i.e. separately for NO2, separately for NOx and separately for PM2.5). We added now “PM2.5” to the method description in the lines 90 and 94.
The authors make note of the very high air pollution in January 2017. I would like the authors to conduct a sensitivity analysis on the influence of this period on their results overall. They should include results of the sensitivity analysis in the discussion, i.e., does this very high exposure period have a strong influence on their findings, or not?
Thank to the Reviewer for the valuable comment. In order to answer the question we conducted a sensitivity analysis for both diseases (bronchitis and asthma exacerbation) in smaller periods (from the 1 January to 31 September) in each year separately (2016 and 2017). Obtained results of calculations are presented below:
Outpatient visits
|
01.01.2016 - 31.08.2016 |
01.01.2017 - 31.08.2017 |
||||||||
|
J00-J21 bronchitis |
J00-J21 bronchitis |
||||||||
|
Parameter |
RR |
LCI |
UCI |
ProbChiSq |
Parameter |
RR |
LCI |
UCI |
ProbChiSq |
|
NO2 |
0.95450 |
0.86710 |
1.05071 |
0.3 |
NO2 |
1.07091 |
1.03474 |
1.10833 |
<.0001 |
|
NO2_3d_avg |
0.93390 |
0.83441 |
1.04525 |
0.2 |
NO2_3d_avg |
1.05021 |
1.01130 |
1.09063 |
0.01 |
|
NO2_5d_avg |
1.07476 |
0.94536 |
1.22187 |
0.2 |
NO2_5d_avg |
1.08428 |
1.03772 |
1.13292 |
0.0003 |
|
NO2_7d_avg |
1.09542 |
0.95064 |
1.26224 |
0.2 |
NO2_7d_avg |
1.11838 |
1.06131 |
1.17852 |
<.0001 |
|
NO2_14d_avg |
1.22039 |
1.01805 |
1.46295 |
0.03 |
NO2_14d_avg |
1.21136 |
1.13350 |
1.29457 |
<.0001 |
|
NO2_30d_avg |
1.68398 |
1.35919 |
2.08637 |
<.0001 |
NO2_30d_avg |
1.21368 |
1.10514 |
1.33287 |
<.0001 |
|
NO2_40d_avg |
1.96825 |
1.55551 |
2.49050 |
<.0001 |
NO2_40d_avg |
1.24209 |
1.11304 |
1.38610 |
0.0001 |
|
NO2_50d_avg |
1.93277 |
1.48007 |
2.52394 |
<.0001 |
NO2_50d_avg |
1.28465 |
1.12700 |
1.46436 |
0.0002 |
|
Parameter |
RR |
LCI |
UCI |
ProbChiSq |
Parameter |
RR |
LCI |
UCI |
ProbChiSq |
|
NOX |
0.95516 |
0.89509 |
1.01926 |
0.1 |
NOX |
1.02789 |
1.00713 |
1.04907 |
0.008 |
|
NOX_3d_avg |
0.90114 |
0.82375 |
0.98579 |
0.02 |
NOX_3d_avg |
1.02037 |
0.99605 |
1.04527 |
0.1 |
|
NOX_5d_avg |
1.00037 |
0.90057 |
1.11123 |
0.9 |
NOX_5d_avg |
1.04280 |
1.01241 |
1.07410 |
0.005 |
|
NOX_7d_avg |
1.01283 |
0.90179 |
1.13755 |
0.8 |
NOX_7d_avg |
1.07302 |
1.03355 |
1.11400 |
0.0002 |
|
NOX_14d_avg |
1.06637 |
0.92115 |
1.23450 |
0.3 |
NOX_14d_avg |
1.19129 |
1.12853 |
1.25755 |
<.0001 |
|
NOX_30d_avg |
1.28748 |
1.09445 |
1.51455 |
0.002 |
NOX_30d_avg |
1.25618 |
1.15912 |
1.36136 |
<.0001 |
|
NOX_40d_avg |
1.43574 |
1.20406 |
1.71200 |
<.0001 |
NOX_40d_avg |
1.28760 |
1.17331 |
1.41303 |
<.0001 |
|
NOX_50d_avg |
1.41301 |
1.16002 |
1.72119 |
0.0006 |
NOX_50d_avg |
1.33881 |
1.19723 |
1.49713 |
<.0001 |
|
Parameter |
RR |
LCI |
UCI |
ProbChiSq |
Parameter |
RR |
LCI |
UCI |
ProbChiSq |
|
PM2.5 |
0.88216 |
0.83170 |
0.93568 |
<.0001 |
PM2.5 |
1.02717 |
1.00753 |
1.04719 |
0.006 |
|
PM2.5_3d_avg |
0.87060 |
0.81530 |
0.92965 |
<.0001 |
PM2.5_3d_avg |
1.02383 |
1.00253 |
1.04558 |
0.02 |
|
PM2.5_5d_avg |
0.88693 |
0.82871 |
0.94923 |
0.0005 |
PM2.5_5d_avg |
1.03267 |
1.00795 |
1.05801 |
0.009 |
|
PM2.5_7d_avg |
0.86922 |
0.80984 |
0.93296 |
0.0001 |
PM2.5_7d_avg |
1.05347 |
1.02302 |
1.08482 |
0.0005 |
|
PM2.5_14d_avg |
0.85535 |
0.79554 |
0.91966 |
<.0001 |
PM2.5_14d_avg |
1.13153 |
1.08763 |
1.17721 |
<.0001 |
|
PM2.5_30d_avg |
0.92838 |
0.86341 |
0.99825 |
0.04 |
PM2.5_30d_avg |
1.16356 |
1.10340 |
1.22701 |
<.0001 |
|
PM2.5_40d_avg |
0.96040 |
0.88949 |
1.03696 |
0.3 |
PM2.5_40d_avg |
1.15563 |
1.08738 |
1.22818 |
<.0001 |
|
PM2.5_50d_avg |
0.95130 |
0.87699 |
1.03190 |
0.2 |
PM2.5_50d_avg |
1.17426 |
1.09589 |
1.25823 |
<.0001 |
|
01.01.2016 - 31.08.2016 |
01.01.2017 - 31.08.2017 |
||||||||
|
J45-J46 asthma exacerbation |
J45-J46 asthma exacerbation |
||||||||
|
Parameter |
RR |
LCI |
UCI |
ProbChiSq |
Parameter |
RR |
LCI |
UCI |
ProbChiSq |
|
NO2 |
1.04600 |
0.98620 |
1.10941 |
0.1 |
NO2 |
1.04399 |
1.00885 |
1.08036 |
0.01 |
|
NO2_3d_avg |
1.03396 |
0.96446 |
1.10847 |
0.3 |
NO2_3d_avg |
1.03390 |
0.99549 |
1.07380 |
0.08 |
|
NO2_5d_avg |
1.09847 |
1.01481 |
1.18903 |
0.02 |
NO2_5d_avg |
1.04236 |
0.99699 |
1.08980 |
0.06 |
|
NO2_7d_avg |
1.12340 |
1.02938 |
1.22602 |
0.009 |
NO2_7d_avg |
1.04873 |
0.99501 |
1.10535 |
0.07 |
|
NO2_14d_avg |
1.22996 |
1.10075 |
1.37435 |
0.0003 |
NO2_14d_avg |
1.09357 |
1.02205 |
1.17010 |
0.009 |
|
NO2_30d_avg |
1.39359 |
1.20536 |
1.61122 |
<.0001 |
NO2_30d_avg |
1.12358 |
1.02397 |
1.23288 |
0.01 |
|
NO2_40d_avg |
1.43869 |
1.21997 |
1.69662 |
<.0001 |
NO2_40d_avg |
1.14006 |
1.02719 |
1.26534 |
0.01 |
|
NO2_50d_avg |
1.45395 |
1.20685 |
1.75165 |
<.0001 |
NO2_50d_avg |
1.17937 |
1.05535 |
1.31797 |
0.003 |
|
Parameter |
RR |
LCI |
UCI |
ProbChiSq |
Parameter |
RR |
LCI |
UCI |
ProbChiSq |
|
NOX |
1.02178 |
0.97950 |
1.06588 |
0.3 |
NOX |
1.02200 |
1.00071 |
1.04374 |
0.04 |
|
NOX_3d_avg |
1.01746 |
0.95969 |
1.07871 |
0.5 |
NOX_3d_avg |
1.02106 |
0.99643 |
1.04629 |
0.09 |
|
NOX_5d_avg |
1.07269 |
1.00249 |
1.14780 |
0.04 |
NOX_5d_avg |
1.02728 |
0.99706 |
1.05842 |
0.07 |
|
NOX_7d_avg |
1.08827 |
1.01082 |
1.17165 |
0.02 |
NOX_7d_avg |
1.03720 |
0.99898 |
1.07689 |
0.05 |
|
NOX_14d_avg |
1.14976 |
1.04670 |
1.26298 |
0.003 |
NOX_14d_avg |
1.08213 |
1.02425 |
1.14329 |
0.004 |
|
NOX_30d_avg |
1.23905 |
1.10672 |
1.38721 |
0.0002 |
NOX_30d_avg |
1.12171 |
1.03410 |
1.21674 |
0.005 |
|
NOX_40d_avg |
1.26027 |
1.11089 |
1.42973 |
0.0003 |
NOX_40d_avg |
1.13813 |
1.03787 |
1.24808 |
0.006 |
|
NOX_50d_avg |
1.26335 |
1.09635 |
1.45579 |
0.001 |
NOX_50d_avg |
1.18814 |
1.07424 |
1.31413 |
0.0008 |
|
Parameter |
RR |
LCI |
UCI |
ProbChiSq |
Parameter |
RR |
LCI |
UCI |
ProbChiSq |
|
PM2.5 |
0.95562 |
0.91723 |
0.99561 |
0.03 |
PM2.5 |
1.01671 |
0.99622 |
1.03762 |
0.1 |
|
PM2.5_3d_avg |
0.94191 |
0.89951 |
0.98632 |
0.01 |
PM2.5_3d_avg |
1.01951 |
0.99789 |
1.04161 |
0.07 |
|
PM2.5_5d_avg |
0.95743 |
0.91198 |
1.00515 |
0.07 |
PM2.5_5d_avg |
1.02197 |
0.99733 |
1.04723 |
0.08 |
|
PM2.5_7d_avg |
0.95525 |
0.90766 |
1.00534 |
0.07 |
PM2.5_7d_avg |
1.02890 |
0.99909 |
1.05961 |
0.05 |
|
PM2.5_14d_avg |
0.96807 |
0.91831 |
1.02053 |
0.2 |
PM2.5_14d_avg |
1.05910 |
1.01732 |
1.10259 |
0.005 |
|
PM2.5_30d_avg |
0.99184 |
0.94038 |
1.04612 |
0.7 |
PM2.5_30d_avg |
1.07965 |
1.02218 |
1.14035 |
0.006 |
|
PM2.5_40d_avg |
0.99457 |
0.94008 |
1.05223 |
0.8 |
PM2.5_40d_avg |
1.09176 |
1.02696 |
1.16065 |
0.004 |
|
PM2.5_50d_avg |
0.98851 |
0.93106 |
1.04951 |
0.7 |
PM2.5_50d_avg |
1.11594 |
1.04716 |
1.18924 |
0.0007 |
Hospitalization
|
01.01.2016 - 31.08.2016 |
01.01.2017 - 31.08.2017 |
||||||||
|
J20-J21 bronchitis |
J20-J21 bronchitis |
||||||||
|
Parameter |
RR |
LCI |
UCI |
ProbChiSq |
Parameter |
RR |
LCI |
UCI |
ProbChiSq |
|
NO2 |
0.99788 |
0.89996 |
1.10645 |
0.9 |
NO2 |
1.06389 |
1.01049 |
1.1201 |
0.01 |
|
NO2_3d_avg |
0.95694 |
0.83930 |
1.09107 |
0.5 |
NO2_3d_avg |
1.06227 |
1.00476 |
1.12308 |
0.03 |
|
NO2_5d_avg |
1.03397 |
0.89007 |
1.20114 |
0.6 |
NO2_5d_avg |
1.10253 |
1.03465 |
1.17487 |
0.002 |
|
NO2_7d_avg |
1.07609 |
0.90932 |
1.27345 |
0.3 |
NO2_7d_avg |
1.13852 |
1.05718 |
1.22612 |
0.0006 |
|
NO2_14d_avg |
1.19694 |
0.97020 |
1.47668 |
0.09 |
NO2_14d_avg |
1.24603 |
1.13382 |
1.36935 |
<.0001 |
|
NO2_30d_avg |
1.92317 |
1.50833 |
2.45211 |
<.0001 |
NO2_30d_avg |
1.44326 |
1.27664 |
1.63163 |
<.0001 |
|
NO2_40d_avg |
2.16994 |
1.66019 |
2.83620 |
<.0001 |
NO2_40d_avg |
1.64926 |
1.42669 |
1.90655 |
<.0001 |
|
NO2_50d_avg |
2.13610 |
1.58063 |
2.88678 |
<.0001 |
NO2_50d_avg |
1.89909 |
1.60477 |
2.24740 |
<.0001 |
|
Parameter |
RR |
LCI |
UCI |
ProbChiSq |
Parameter |
RR |
LCI |
UCI |
ProbChiSq |
|
NOX |
0.97252 |
0.90139 |
1.04925 |
0.4 |
NOX |
1.01805 |
0.98571 |
1.0515 |
0.2 |
|
NOX_3d_avg |
0.92286 |
0.82998 |
1.02613 |
0.1 |
NOX_3d_avg |
1.02627 |
0.98942 |
1.0645 |
0.1 |
|
NOX_5d_avg |
0.97021 |
0.85870 |
1.09620 |
0.6 |
NOX_5d_avg |
1.05154 |
1.00692 |
1.0981 |
0.02 |
|
NOX_7d_avg |
0.99683 |
0.86912 |
1.14331 |
0.9 |
NOX_7d_avg |
1.07916 |
1.02321 |
1.13817 |
0.005 |
|
NOX_14d_avg |
1.07388 |
0.90841 |
1.26949 |
0.4 |
NOX_14d_avg |
1.16721 |
1.07974 |
1.26177 |
0.0001 |
|
NOX_30d_avg |
1.43023 |
1.18956 |
1.71959 |
0.0001 |
NOX_30d_avg |
1.36009 |
1.22064 |
1.51548 |
<.0001 |
|
NOX_40d_avg |
1.58316 |
1.29760 |
1.93156 |
<.0001 |
NOX_40d_avg |
1.56561 |
1.38007 |
1.77610 |
<.0001 |
|
NOX_50d_avg |
1.56523 |
1.25349 |
1.95450 |
<.0001 |
NOX_50d_avg |
1.79830 |
1.55878 |
2.07463 |
<.0001 |
|
Parameter |
RR |
LCI |
UCI |
ProbChiSq |
Parameter |
RR |
LCI |
UCI |
ProbChiSq |
|
PM2.5 |
0.91154 |
0.85014 |
0.97738 |
0.009 |
PM2.5 |
0.99422 |
0.96206 |
1.0274 |
0.7 |
|
PM2.5_3d_avg |
0.85947 |
0.79196 |
0.93273 |
0.0003 |
PM2.5_3d_avg |
1.00798 |
0.97470 |
1.0424 |
0.6 |
|
PM2.5_5d_avg |
0.85996 |
0.79169 |
0.93412 |
0.0004 |
PM2.5_5d_avg |
1.02799 |
0.99071 |
1.0667 |
0.1 |
|
PM2.5_7d_avg |
0.85948 |
0.79018 |
0.93486 |
0.0004 |
PM2.5_7d_avg |
1.05020 |
1.00602 |
1.0963 |
0.02 |
|
PM2.5_14d_avg |
0.85747 |
0.78574 |
0.93575 |
0.0006 |
PM2.5_14d_avg |
1.12088 |
1.05844 |
1.18701 |
<.0001 |
|
PM2.5_30d_avg |
0.96142 |
0.88196 |
1.04804 |
0.3 |
PM2.5_30d_avg |
1.23653 |
1.15065 |
1.32882 |
<.0001 |
|
PM2.5_40d_avg |
1.00879 |
0.92212 |
1.10361 |
0.8 |
PM2.5_40d_avg |
1.32060 |
1.21555 |
1.43474 |
<.0001 |
|
PM2.5_50d_avg |
1.00370 |
0.91193 |
1.10472 |
0.9 |
PM2.5_50d_avg |
1.42486 |
1.30206 |
1.55924 |
<.0001 |
|
01.01.2016 - 31.08.2016 |
01.01.2017 - 31.08.2017 |
||||||||
|
J45-J46 asthma exacerbation |
J45-J46 asthma exacerbation |
||||||||
|
Parameter |
RR |
LCI |
UCI |
ProbChiSq |
Parameter |
RR |
LCI |
UCI |
ProbChiSq |
|
NO2 |
1.19810 |
1.06176 |
1.35195 |
0.003 |
NO2 |
1.17584 |
1.09484 |
1.26284 |
<.0001 |
|
NO2_3d_avg |
1.03613 |
0.88702 |
1.21031 |
0.6 |
NO2_3d_avg |
1.08847 |
1.00113 |
1.18342 |
0.04 |
|
NO2_5d_avg |
1.03861 |
0.86824 |
1.24240 |
0.6 |
NO2_5d_avg |
1.06606 |
0.96640 |
1.17600 |
0.2 |
|
NO2_7d_avg |
1.16966 |
0.95976 |
1.42546 |
0.1 |
NO2_7d_avg |
1.06829 |
0.95426 |
1.19594 |
0.2 |
|
NO2_14d_avg |
1.22779 |
0.95579 |
1.57719 |
0.1 |
NO2_14d_avg |
1.09790 |
0.94851 |
1.27081 |
0.2 |
|
NO2_30d_avg |
1.49175 |
1.07646 |
2.06727 |
0.01 |
NO2_30d_avg |
1.06858 |
0.87786 |
1.30074 |
0.5 |
|
NO2_40d_avg |
1.53850 |
1.06103 |
2.23084 |
0.02 |
NO2_40d_avg |
1.02336 |
0.81274 |
1.28856 |
0.8 |
|
NO2_50d_avg |
1.73990 |
1.14757 |
2.63795 |
0.009 |
NO2_50d_avg |
1.10616 |
0.86177 |
1.41985 |
0.4 |
|
Parameter |
RR |
LCI |
UCI |
ProbChiSq |
Parameter |
RR |
LCI |
UCI |
ProbChiSq |
|
NOX |
1.09634 |
0.99848 |
1.20380 |
0.05 |
NOX |
1.07411 |
1.02758 |
1.12276 |
0.001 |
|
NOX_3d_avg |
0.99345 |
0.86921 |
1.13544 |
0.9 |
NOX_3d_avg |
1.05365 |
0.99788 |
1.11254 |
0.05 |
|
NOX_5d_avg |
1.00229 |
0.85955 |
1.16872 |
0.9 |
NOX_5d_avg |
1.04859 |
0.98089 |
1.12098 |
0.1 |
|
NOX_7d_avg |
1.09867 |
0.92824 |
1.30040 |
0.2 |
NOX_7d_avg |
1.05352 |
0.97132 |
1.14268 |
0.2 |
|
NOX_14d_avg |
1.11611 |
0.90414 |
1.37778 |
0.3 |
NOX_14d_avg |
1.10606 |
0.98179 |
1.24606 |
0.09 |
|
NOX_30d_avg |
1.27086 |
0.98371 |
1.64181 |
0.06 |
NOX_30d_avg |
1.09555 |
0.92482 |
1.29781 |
0.2 |
|
NOX_40d_avg |
1.28548 |
0.96688 |
1.70906 |
0.08 |
NOX_40d_avg |
1.05562 |
0.86343 |
1.29059 |
0.5 |
|
NOX_50d_avg |
1.38598 |
1.00855 |
1.90466 |
0.04 |
NOX_50d_avg |
1.13350 |
0.90633 |
1.41761 |
0.2 |
|
Parameter |
RR |
LCI |
UCI |
ProbChiSq |
Parameter |
RR |
LCI |
UCI |
ProbChiSq |
|
PM2.5 |
0.91545 |
0.82790 |
1.01225 |
0.08 |
PM2.5 |
1.04030 |
0.99435 |
1.08836 |
0.08 |
|
PM2.5_3d_avg |
0.87654 |
0.78133 |
0.98336 |
0.02 |
PM2.5_3d_avg |
1.03959 |
0.99005 |
1.09160 |
0.1 |
|
PM2.5_5d_avg |
0.91363 |
0.81341 |
1.02618 |
0.1 |
PM2.5_5d_avg |
1.03727 |
0.98108 |
1.09667 |
0.1 |
|
PM2.5_7d_avg |
0.93697 |
0.83295 |
1.05398 |
0.2 |
PM2.5_7d_avg |
1.03813 |
0.97240 |
1.10829 |
0.2 |
|
PM2.5_14d_avg |
0.93115 |
0.82250 |
1.05416 |
0.2 |
PM2.5_14d_avg |
1.06997 |
0.97974 |
1.16850 |
0.1 |
|
PM2.5_30d_avg |
0.97190 |
0.85751 |
1.10156 |
0.6 |
PM2.5_30d_avg |
1.05141 |
0.93754 |
1.17912 |
0.3 |
|
PM2.5_40d_avg |
0.98353 |
0.86246 |
1.12158 |
0.8 |
PM2.5_40d_avg |
1.03804 |
0.90831 |
1.18631 |
0.5 |
|
PM2.5_50d_avg |
0.98870 |
0.85953 |
1.13729 |
0.8 |
PM2.5_50d_avg |
1.06007 |
0.92017 |
1.22125 |
0.4 |
We hope that the current version of the presented data and our explanation in the Discussion section could be accepted by the Reviewer. We added the following sentences (kindly see the line 214-225).
The more, so that the observation period is quite long and there was a visible deterioration in the quality of ambient air between 7 and 11 January 2017 we decided to reassess the RR value for two separate periods (from January 1 to August 30 of each year), respectively. Obtained results show some differences in the picture of dose-response for each year (detailed results are available upon request). The relationship between outpatient visits due to bronchitis and asthma exacerbation and NO2 or NOx concentration was similar in 2017 and in whole studied period. Additionally, a statistically significant effect of fine particle on outpatient visits was revealed, including all moving average concentration of pollution. On the second hand, the lower concentrations in 2016 resulted in significant reduction of RR value. In the case of hospitalization, values of RR were not statistically significant in longer time exposure than this, expressed by 1 or 3 day- moving average concentration. In conclusion, a shorter time period results in fewer recorded health events, which can lead to changes in RR values.
Presentation issues, language and typos:
Use of subscripts for pollutant abbreviations and units of measurement should be made consistent in text and tables.
Done in total manuscript body.
Lines 31-32: further describe improvement in air quality. Are conditions in Europe improving on average?
Thank you for the comment. To explain the situation we cited the opinion published by EEA (available at the website: https://www.eea.europa.eu/data-and-maps/indicators/eea-32-nitrogen-oxides-nox-emissions-1/assessment.2010-08-19.0140149032-3). Despite the observed decline in NOx emissions in Europe (mostly in the road transport), concentrations remain high in many countries due to the burning of fossil fuels for energy and heating purposes. We believe that such explanation will be satisfied with the Reviewer.
Line 32: Substitute "...is steadily improving" in place of "is systematically improving"
Done, kindly see the current version
Line 48: "...energy production are the most..."
Done, kindly see the current version
Line 56: "...health effects depending..."
Done, kindly see the current version
Line 64: Word choice, suggest "crowded cities" rather than "crowded agglomerations"
Done, kindly see the current version
Line 69: If the data cover a time period of 2016-2017, why is "(in 2015)" included here? This is not clear.
Thank you for this remark, we agree with You. Finally, we decided to remove the year (in a bracket). Moreover, we explain, that the number of inhabitants in the studied area in particular years 2015, 2016 and 2017 was respectively: 2,512,449 and 2,498,850 and 2486,155 (data available on the Statistics Poland - Local Data Bank in Warsaw: https://bdl.stat.gov.pl/BDL/dane/podgrup/temat)
Line 73: Substitute "including" instead of "occurring both"
Done, kindly see the current version
Line 76: There should be a break in text after a table, recommend inserting 1-2 blank lines.
Done, kindly see the current version
Line 79: "Air quality" is more commonly used than "aerosanitary" (also line 107)
Done, kindly see the current version (now, line 82 and 109)
Line 86: Insert space after the table.
Done, kindly see the current version
Line 94: Delete "a" before season
Done, kindly see the current version
Line 117: Insert "a" before "significant"
Done, kindly see the current version
Line 120: delete "function"
Done, kindly see the current version
Line 123: This is Table 3. Suggest presenting results here with two decimal places to improve readability.
We propose to leave the current version with three decimal places, we hope that the technical edition will be enough clear
Table 3, section on asthma exacerbation, NO2, 3day lag hospitalizations: there is a typo, zero instead of dash
Done, kindly see the current version
Line 173: Substitute "has improved" in place of "was going better"
Done, kindly see the current version
Line 184: Insert a period after "production"
Done, kindly see the current version
Line 191: Use "giving" rather than "gives"
Done, kindly see the current version
Line 196: Insert "the" before "youngest"
Done, kindly see the current version
Line 198: "Admissions"
We don’t understand why and what we have to do? We propose to leave a previous version ‘…hospital admission…’ (now in line 206)
Line 199: "The number of daily bronchitis visits and hospitalizations was higher and the..."
Done, kindly see the current version
Reviewer 3 Report
This study has evaluated the effect of gases and fine particles in the ambient air and daily bronchitis and asthma exacerbation in Silesian Voivodeship in Poland. The study is well conducted and I have a few comments.
You have analyzed the fine particles as well. Why the title is only NOx and NO2?
Present the quantitative results (relative risk value) in the Abstract. Currently it is only description and interpretation.
Why did you choose 60-day moving average? In a time series analysis it may compete with seasonality. I have seen a recent paper ( https://doi.org/10.1016/j.envpol.2019.113121 ) that analyzed up to 45 days but even that one could be very long period for short-term studies.
How did you adjust for season? which spline did you use? which degree of freedom?
How the temperature was controlled in your analysis? Was it the same lag? a cross-basis? etc? Also for other co-variates. Please clarify.
Did you have data to evaluate effect modification by some covariates? e.g. gender?
The sensitivity analysis is missing? Have you done any?
Author Response
Rewiever 3
You have analyzed the fine particles as well. Why the title is only NOx and NO2?
Thank you for this remark, we have intentionally left only nitric oxide concentration in the title because in our previous work we detailed described relationship between PM10 or PM2.5 concentrations and respiratory health problems (Kowalska M, Skrzypek M, Kowalski M, Niewiadomska E, Czech E, Cyrys J. The relationship between daily concentration of fine particulate matter in ambient air and exacerbation of respiratory diseases in Silesian Agglomeration, Poland. Int. J. Environ. Res. Public Health 2019, 16, 1131; doi:10.3390/ijerph16071131)
Present the quantitative results (relative risk value) in the Abstract. Currently it is only description and interpretation.
Thank you for this comment, and with your expectation, we rearranged somewhat the following sentences (line 28-29): ‘The strongest relationship was observed in the case of NO2 and outpatient visits due to bronchitis, e.g. RR=1.434 (1.308-1.571) for exposure expressed by the 50 day moving average concentration.’
Why did you choose a 60-day moving average? In a time series analysis it may compete with seasonality. I have seen a recent paper ( https://doi.org/10.1016/j.envpol.2019.113121 ) that analyzed up to 45 days but even that one could be very long period for short-term studies.
Thank you for the remark, you are right. We removed the 60-day moving average concentration from the analysis and consequently we correct data in Table 3 and on Figures 2-4. We believe that such modification will be accepted by the Reviewer.
How did you adjust for season? which spline did you use? which degree of freedom?
In the Poisson regression model, we just introduced a categorical variable that represented a season of the year.
How the temperature was controlled in your analysis? Was it the same lag? a cross-basis? etc? Also for other co-variates. Please clarify.
The average ambient air temperature, as well as atmospheric pressure and relative humidity, were taken on the day of hospital admission or outpatient visit. Moreover, we decided to some modification of the Discussion, we point some weakness of the used model in the following sentence:
Finally, we can't forgot, that usually environmental stressors show effects delayed in time, and dose-response analysis require proper statistical model. Our analysis was based on the multivariable log-linear Poisson regression model, which seems too simple for some epidemiologists. Promising proposition for the future analyses could be a new more flexible variants of distributed lag non-linear models (DLNM) implemented in the statistical environment R [22].
Did you have data to evaluate effect modification by some covariates? e.g. gender?
Thank you for this comment, but your suggestion is impossible to realize because we don’t have data on gender.
The sensitivity analysis is missing? Have you done any?
Thank you for this remark, we explain that some type of sensitivity analysis was recalculation of RR value in two shorter periods, separately from 01 January to 31 September 2016 and 01 January to 31 September 2017. Obtained results are presented below:
Outpatient visits
|
01.01.2016 - 31.08.2016 |
01.01.2017 - 31.08.2017 |
||||||||
|
J00-J21 bronchitis |
J00-J21 bronchitis |
||||||||
|
Parameter |
RR |
LCI |
UCI |
ProbChiSq |
Parameter |
RR |
LCI |
UCI |
ProbChiSq |
|
NO2 |
0.95450 |
0.86710 |
1.05071 |
0.3 |
NO2 |
1.07091 |
1.03474 |
1.10833 |
<.0001 |
|
NO2_3d_avg |
0.93390 |
0.83441 |
1.04525 |
0.2 |
NO2_3d_avg |
1.05021 |
1.01130 |
1.09063 |
0.01 |
|
NO2_5d_avg |
1.07476 |
0.94536 |
1.22187 |
0.2 |
NO2_5d_avg |
1.08428 |
1.03772 |
1.13292 |
0.0003 |
|
NO2_7d_avg |
1.09542 |
0.95064 |
1.26224 |
0.2 |
NO2_7d_avg |
1.11838 |
1.06131 |
1.17852 |
<.0001 |
|
NO2_14d_avg |
1.22039 |
1.01805 |
1.46295 |
0.03 |
NO2_14d_avg |
1.21136 |
1.13350 |
1.29457 |
<.0001 |
|
NO2_30d_avg |
1.68398 |
1.35919 |
2.08637 |
<.0001 |
NO2_30d_avg |
1.21368 |
1.10514 |
1.33287 |
<.0001 |
|
NO2_40d_avg |
1.96825 |
1.55551 |
2.49050 |
<.0001 |
NO2_40d_avg |
1.24209 |
1.11304 |
1.38610 |
0.0001 |
|
NO2_50d_avg |
1.93277 |
1.48007 |
2.52394 |
<.0001 |
NO2_50d_avg |
1.28465 |
1.12700 |
1.46436 |
0.0002 |
|
Parameter |
RR |
LCI |
UCI |
ProbChiSq |
Parameter |
RR |
LCI |
UCI |
ProbChiSq |
|
NOX |
0.95516 |
0.89509 |
1.01926 |
0.1 |
NOX |
1.02789 |
1.00713 |
1.04907 |
0.008 |
|
NOX_3d_avg |
0.90114 |
0.82375 |
0.98579 |
0.02 |
NOX_3d_avg |
1.02037 |
0.99605 |
1.04527 |
0.1 |
|
NOX_5d_avg |
1.00037 |
0.90057 |
1.11123 |
0.9 |
NOX_5d_avg |
1.04280 |
1.01241 |
1.07410 |
0.005 |
|
NOX_7d_avg |
1.01283 |
0.90179 |
1.13755 |
0.8 |
NOX_7d_avg |
1.07302 |
1.03355 |
1.11400 |
0.0002 |
|
NOX_14d_avg |
1.06637 |
0.92115 |
1.23450 |
0.3 |
NOX_14d_avg |
1.19129 |
1.12853 |
1.25755 |
<.0001 |
|
NOX_30d_avg |
1.28748 |
1.09445 |
1.51455 |
0.002 |
NOX_30d_avg |
1.25618 |
1.15912 |
1.36136 |
<.0001 |
|
NOX_40d_avg |
1.43574 |
1.20406 |
1.71200 |
<.0001 |
NOX_40d_avg |
1.28760 |
1.17331 |
1.41303 |
<.0001 |
|
NOX_50d_avg |
1.41301 |
1.16002 |
1.72119 |
0.0006 |
NOX_50d_avg |
1.33881 |
1.19723 |
1.49713 |
<.0001 |
|
Parameter |
RR |
LCI |
UCI |
ProbChiSq |
Parameter |
RR |
LCI |
UCI |
ProbChiSq |
|
PM2.5 |
0.88216 |
0.83170 |
0.93568 |
<.0001 |
PM2.5 |
1.02717 |
1.00753 |
1.04719 |
0.006 |
|
PM2.5_3d_avg |
0.87060 |
0.81530 |
0.92965 |
<.0001 |
PM2.5_3d_avg |
1.02383 |
1.00253 |
1.04558 |
0.02 |
|
PM2.5_5d_avg |
0.88693 |
0.82871 |
0.94923 |
0.0005 |
PM2.5_5d_avg |
1.03267 |
1.00795 |
1.05801 |
0.009 |
|
PM2.5_7d_avg |
0.86922 |
0.80984 |
0.93296 |
0.0001 |
PM2.5_7d_avg |
1.05347 |
1.02302 |
1.08482 |
0.0005 |
|
PM2.5_14d_avg |
0.85535 |
0.79554 |
0.91966 |
<.0001 |
PM2.5_14d_avg |
1.13153 |
1.08763 |
1.17721 |
<.0001 |
|
PM2.5_30d_avg |
0.92838 |
0.86341 |
0.99825 |
0.04 |
PM2.5_30d_avg |
1.16356 |
1.10340 |
1.22701 |
<.0001 |
|
PM2.5_40d_avg |
0.96040 |
0.88949 |
1.03696 |
0.3 |
PM2.5_40d_avg |
1.15563 |
1.08738 |
1.22818 |
<.0001 |
|
PM2.5_50d_avg |
0.95130 |
0.87699 |
1.03190 |
0.2 |
PM2.5_50d_avg |
1.17426 |
1.09589 |
1.25823 |
<.0001 |
|
01.01.2016 - 31.08.2016 |
01.01.2017 - 31.08.2017 |
||||||||
|
J45-J46 asthma exacerbation |
J45-J46 asthma exacerbation |
||||||||
|
Parameter |
RR |
LCI |
UCI |
ProbChiSq |
Parameter |
RR |
LCI |
UCI |
ProbChiSq |
|
NO2 |
1.04600 |
0.98620 |
1.10941 |
0.1 |
NO2 |
1.04399 |
1.00885 |
1.08036 |
0.01 |
|
NO2_3d_avg |
1.03396 |
0.96446 |
1.10847 |
0.3 |
NO2_3d_avg |
1.03390 |
0.99549 |
1.07380 |
0.08 |
|
NO2_5d_avg |
1.09847 |
1.01481 |
1.18903 |
0.02 |
NO2_5d_avg |
1.04236 |
0.99699 |
1.08980 |
0.06 |
|
NO2_7d_avg |
1.12340 |
1.02938 |
1.22602 |
0.009 |
NO2_7d_avg |
1.04873 |
0.99501 |
1.10535 |
0.07 |
|
NO2_14d_avg |
1.22996 |
1.10075 |
1.37435 |
0.0003 |
NO2_14d_avg |
1.09357 |
1.02205 |
1.17010 |
0.009 |
|
NO2_30d_avg |
1.39359 |
1.20536 |
1.61122 |
<.0001 |
NO2_30d_avg |
1.12358 |
1.02397 |
1.23288 |
0.01 |
|
NO2_40d_avg |
1.43869 |
1.21997 |
1.69662 |
<.0001 |
NO2_40d_avg |
1.14006 |
1.02719 |
1.26534 |
0.01 |
|
NO2_50d_avg |
1.45395 |
1.20685 |
1.75165 |
<.0001 |
NO2_50d_avg |
1.17937 |
1.05535 |
1.31797 |
0.003 |
|
Parameter |
RR |
LCI |
UCI |
ProbChiSq |
Parameter |
RR |
LCI |
UCI |
ProbChiSq |
|
NOX |
1.02178 |
0.97950 |
1.06588 |
0.3 |
NOX |
1.02200 |
1.00071 |
1.04374 |
0.04 |
|
NOX_3d_avg |
1.01746 |
0.95969 |
1.07871 |
0.5 |
NOX_3d_avg |
1.02106 |
0.99643 |
1.04629 |
0.09 |
|
NOX_5d_avg |
1.07269 |
1.00249 |
1.14780 |
0.04 |
NOX_5d_avg |
1.02728 |
0.99706 |
1.05842 |
0.07 |
|
NOX_7d_avg |
1.08827 |
1.01082 |
1.17165 |
0.02 |
NOX_7d_avg |
1.03720 |
0.99898 |
1.07689 |
0.05 |
|
NOX_14d_avg |
1.14976 |
1.04670 |
1.26298 |
0.003 |
NOX_14d_avg |
1.08213 |
1.02425 |
1.14329 |
0.004 |
|
NOX_30d_avg |
1.23905 |
1.10672 |
1.38721 |
0.0002 |
NOX_30d_avg |
1.12171 |
1.03410 |
1.21674 |
0.005 |
|
NOX_40d_avg |
1.26027 |
1.11089 |
1.42973 |
0.0003 |
NOX_40d_avg |
1.13813 |
1.03787 |
1.24808 |
0.006 |
|
NOX_50d_avg |
1.26335 |
1.09635 |
1.45579 |
0.001 |
NOX_50d_avg |
1.18814 |
1.07424 |
1.31413 |
0.0008 |
|
Parameter |
RR |
LCI |
UCI |
ProbChiSq |
Parameter |
RR |
LCI |
UCI |
ProbChiSq |
|
PM2.5 |
0.95562 |
0.91723 |
0.99561 |
0.03 |
PM2.5 |
1.01671 |
0.99622 |
1.03762 |
0.1 |
|
PM2.5_3d_avg |
0.94191 |
0.89951 |
0.98632 |
0.01 |
PM2.5_3d_avg |
1.01951 |
0.99789 |
1.04161 |
0.07 |
|
PM2.5_5d_avg |
0.95743 |
0.91198 |
1.00515 |
0.07 |
PM2.5_5d_avg |
1.02197 |
0.99733 |
1.04723 |
0.08 |
|
PM2.5_7d_avg |
0.95525 |
0.90766 |
1.00534 |
0.07 |
PM2.5_7d_avg |
1.02890 |
0.99909 |
1.05961 |
0.05 |
|
PM2.5_14d_avg |
0.96807 |
0.91831 |
1.02053 |
0.2 |
PM2.5_14d_avg |
1.05910 |
1.01732 |
1.10259 |
0.005 |
|
PM2.5_30d_avg |
0.99184 |
0.94038 |
1.04612 |
0.7 |
PM2.5_30d_avg |
1.07965 |
1.02218 |
1.14035 |
0.006 |
|
PM2.5_40d_avg |
0.99457 |
0.94008 |
1.05223 |
0.8 |
PM2.5_40d_avg |
1.09176 |
1.02696 |
1.16065 |
0.004 |
|
PM2.5_50d_avg |
0.98851 |
0.93106 |
1.04951 |
0.7 |
PM2.5_50d_avg |
1.11594 |
1.04716 |
1.18924 |
0.0007 |
Hospitalization
|
01.01.2016 - 31.08.2016 |
01.01.2017 - 31.08.2017 |
||||||||
|
J20-J21 bronchitis |
J20-J21 bronchitis |
||||||||
|
Parameter |
RR |
LCI |
UCI |
ProbChiSq |
Parameter |
RR |
LCI |
UCI |
ProbChiSq |
|
NO2 |
0.99788 |
0.89996 |
1.10645 |
0.9 |
NO2 |
1.06389 |
1.01049 |
1.1201 |
0.01 |
|
NO2_3d_avg |
0.95694 |
0.83930 |
1.09107 |
0.5 |
NO2_3d_avg |
1.06227 |
1.00476 |
1.12308 |
0.03 |
|
NO2_5d_avg |
1.03397 |
0.89007 |
1.20114 |
0.6 |
NO2_5d_avg |
1.10253 |
1.03465 |
1.17487 |
0.002 |
|
NO2_7d_avg |
1.07609 |
0.90932 |
1.27345 |
0.3 |
NO2_7d_avg |
1.13852 |
1.05718 |
1.22612 |
0.0006 |
|
NO2_14d_avg |
1.19694 |
0.97020 |
1.47668 |
0.09 |
NO2_14d_avg |
1.24603 |
1.13382 |
1.36935 |
<.0001 |
|
NO2_30d_avg |
1.92317 |
1.50833 |
2.45211 |
<.0001 |
NO2_30d_avg |
1.44326 |
1.27664 |
1.63163 |
<.0001 |
|
NO2_40d_avg |
2.16994 |
1.66019 |
2.83620 |
<.0001 |
NO2_40d_avg |
1.64926 |
1.42669 |
1.90655 |
<.0001 |
|
NO2_50d_avg |
2.13610 |
1.58063 |
2.88678 |
<.0001 |
NO2_50d_avg |
1.89909 |
1.60477 |
2.24740 |
<.0001 |
|
Parameter |
RR |
LCI |
UCI |
ProbChiSq |
Parameter |
RR |
LCI |
UCI |
ProbChiSq |
|
NOX |
0.97252 |
0.90139 |
1.04925 |
0.4 |
NOX |
1.01805 |
0.98571 |
1.0515 |
0.2 |
|
NOX_3d_avg |
0.92286 |
0.82998 |
1.02613 |
0.1 |
NOX_3d_avg |
1.02627 |
0.98942 |
1.0645 |
0.1 |
|
NOX_5d_avg |
0.97021 |
0.85870 |
1.09620 |
0.6 |
NOX_5d_avg |
1.05154 |
1.00692 |
1.0981 |
0.02 |
|
NOX_7d_avg |
0.99683 |
0.86912 |
1.14331 |
0.9 |
NOX_7d_avg |
1.07916 |
1.02321 |
1.13817 |
0.005 |
|
NOX_14d_avg |
1.07388 |
0.90841 |
1.26949 |
0.4 |
NOX_14d_avg |
1.16721 |
1.07974 |
1.26177 |
0.0001 |
|
NOX_30d_avg |
1.43023 |
1.18956 |
1.71959 |
0.0001 |
NOX_30d_avg |
1.36009 |
1.22064 |
1.51548 |
<.0001 |
|
NOX_40d_avg |
1.58316 |
1.29760 |
1.93156 |
<.0001 |
NOX_40d_avg |
1.56561 |
1.38007 |
1.77610 |
<.0001 |
|
NOX_50d_avg |
1.56523 |
1.25349 |
1.95450 |
<.0001 |
NOX_50d_avg |
1.79830 |
1.55878 |
2.07463 |
<.0001 |
|
Parameter |
RR |
LCI |
UCI |
ProbChiSq |
Parameter |
RR |
LCI |
UCI |
ProbChiSq |
|
PM2.5 |
0.91154 |
0.85014 |
0.97738 |
0.009 |
PM2.5 |
0.99422 |
0.96206 |
1.0274 |
0.7 |
|
PM2.5_3d_avg |
0.85947 |
0.79196 |
0.93273 |
0.0003 |
PM2.5_3d_avg |
1.00798 |
0.97470 |
1.0424 |
0.6 |
|
PM2.5_5d_avg |
0.85996 |
0.79169 |
0.93412 |
0.0004 |
PM2.5_5d_avg |
1.02799 |
0.99071 |
1.0667 |
0.1 |
|
PM2.5_7d_avg |
0.85948 |
0.79018 |
0.93486 |
0.0004 |
PM2.5_7d_avg |
1.05020 |
1.00602 |
1.0963 |
0.02 |
|
PM2.5_14d_avg |
0.85747 |
0.78574 |
0.93575 |
0.0006 |
PM2.5_14d_avg |
1.12088 |
1.05844 |
1.18701 |
<.0001 |
|
PM2.5_30d_avg |
0.96142 |
0.88196 |
1.04804 |
0.3 |
PM2.5_30d_avg |
1.23653 |
1.15065 |
1.32882 |
<.0001 |
|
PM2.5_40d_avg |
1.00879 |
0.92212 |
1.10361 |
0.8 |
PM2.5_40d_avg |
1.32060 |
1.21555 |
1.43474 |
<.0001 |
|
PM2.5_50d_avg |
1.00370 |
0.91193 |
1.10472 |
0.9 |
PM2.5_50d_avg |
1.42486 |
1.30206 |
1.55924 |
<.0001 |
|
01.01.2016 - 31.08.2016 |
01.01.2017 - 31.08.2017 |
||||||||
|
J45-J46 asthma exacerbation |
J45-J46 asthma exacerbation |
||||||||
|
Parameter |
RR |
LCI |
UCI |
ProbChiSq |
Parameter |
RR |
LCI |
UCI |
ProbChiSq |
|
NO2 |
1.19810 |
1.06176 |
1.35195 |
0.003 |
NO2 |
1.17584 |
1.09484 |
1.26284 |
<.0001 |
|
NO2_3d_avg |
1.03613 |
0.88702 |
1.21031 |
0.6 |
NO2_3d_avg |
1.08847 |
1.00113 |
1.18342 |
0.04 |
|
NO2_5d_avg |
1.03861 |
0.86824 |
1.24240 |
0.6 |
NO2_5d_avg |
1.06606 |
0.96640 |
1.17600 |
0.2 |
|
NO2_7d_avg |
1.16966 |
0.95976 |
1.42546 |
0.1 |
NO2_7d_avg |
1.06829 |
0.95426 |
1.19594 |
0.2 |
|
NO2_14d_avg |
1.22779 |
0.95579 |
1.57719 |
0.1 |
NO2_14d_avg |
1.09790 |
0.94851 |
1.27081 |
0.2 |
|
NO2_30d_avg |
1.49175 |
1.07646 |
2.06727 |
0.01 |
NO2_30d_avg |
1.06858 |
0.87786 |
1.30074 |
0.5 |
|
NO2_40d_avg |
1.53850 |
1.06103 |
2.23084 |
0.02 |
NO2_40d_avg |
1.02336 |
0.81274 |
1.28856 |
0.8 |
|
NO2_50d_avg |
1.73990 |
1.14757 |
2.63795 |
0.009 |
NO2_50d_avg |
1.10616 |
0.86177 |
1.41985 |
0.4 |
|
Parameter |
RR |
LCI |
UCI |
ProbChiSq |
Parameter |
RR |
LCI |
UCI |
ProbChiSq |
|
NOX |
1.09634 |
0.99848 |
1.20380 |
0.05 |
NOX |
1.07411 |
1.02758 |
1.12276 |
0.001 |
|
NOX_3d_avg |
0.99345 |
0.86921 |
1.13544 |
0.9 |
NOX_3d_avg |
1.05365 |
0.99788 |
1.11254 |
0.05 |
|
NOX_5d_avg |
1.00229 |
0.85955 |
1.16872 |
0.9 |
NOX_5d_avg |
1.04859 |
0.98089 |
1.12098 |
0.1 |
|
NOX_7d_avg |
1.09867 |
0.92824 |
1.30040 |
0.2 |
NOX_7d_avg |
1.05352 |
0.97132 |
1.14268 |
0.2 |
|
NOX_14d_avg |
1.11611 |
0.90414 |
1.37778 |
0.3 |
NOX_14d_avg |
1.10606 |
0.98179 |
1.24606 |
0.09 |
|
NOX_30d_avg |
1.27086 |
0.98371 |
1.64181 |
0.06 |
NOX_30d_avg |
1.09555 |
0.92482 |
1.29781 |
0.2 |
|
NOX_40d_avg |
1.28548 |
0.96688 |
1.70906 |
0.08 |
NOX_40d_avg |
1.05562 |
0.86343 |
1.29059 |
0.5 |
|
NOX_50d_avg |
1.38598 |
1.00855 |
1.90466 |
0.04 |
NOX_50d_avg |
1.13350 |
0.90633 |
1.41761 |
0.2 |
|
Parameter |
RR |
LCI |
UCI |
ProbChiSq |
Parameter |
RR |
LCI |
UCI |
ProbChiSq |
|
PM2.5 |
0.91545 |
0.82790 |
1.01225 |
0.08 |
PM2.5 |
1.04030 |
0.99435 |
1.08836 |
0.08 |
|
PM2.5_3d_avg |
0.87654 |
0.78133 |
0.98336 |
0.02 |
PM2.5_3d_avg |
1.03959 |
0.99005 |
1.09160 |
0.1 |
|
PM2.5_5d_avg |
0.91363 |
0.81341 |
1.02618 |
0.1 |
PM2.5_5d_avg |
1.03727 |
0.98108 |
1.09667 |
0.1 |
|
PM2.5_7d_avg |
0.93697 |
0.83295 |
1.05398 |
0.2 |
PM2.5_7d_avg |
1.03813 |
0.97240 |
1.10829 |
0.2 |
|
PM2.5_14d_avg |
0.93115 |
0.82250 |
1.05416 |
0.2 |
PM2.5_14d_avg |
1.06997 |
0.97974 |
1.16850 |
0.1 |
|
PM2.5_30d_avg |
0.97190 |
0.85751 |
1.10156 |
0.6 |
PM2.5_30d_avg |
1.05141 |
0.93754 |
1.17912 |
0.3 |
|
PM2.5_40d_avg |
0.98353 |
0.86246 |
1.12158 |
0.8 |
PM2.5_40d_avg |
1.03804 |
0.90831 |
1.18631 |
0.5 |
|
PM2.5_50d_avg |
0.98870 |
0.85953 |
1.13729 |
0.8 |
PM2.5_50d_avg |
1.06007 |
0.92017 |
1.22125 |
0.4 |
We hope that the current version of the presented data and our explanation in the Discussion section could be accepted by the Reviewer. We added the following sentences (kindly see the line 214-225).
The more, so that the observation period is quite long and there was a visible deterioration in the quality of ambient air between 7 and 11 January 2017 we decided to reassess the RR value for two separate periods (from January 1 to August 30 of each year), respectively. Obtained results show some differences in the picture of dose-response for each year (detailed results are available upon request). The relationship between outpatient visits due to bronchitis and asthma exacerbation and NO2 or NOx concentration was similar in 2017 and in whole studied period. Additionally, a statistically significant effect of fine particle on outpatient visits was revealed, including all moving average concentration of pollution. On the second hand, the lower concentrations in 2016 resulted in significant reduction of RR value. In the case of hospitalization, values of RR were not statistically significant in longer time exposure than this, expressed by 1 or 3 day- moving average concentration. In conclusion, a shorter time period results in fewer recorded health events, which can lead to changes in RR values.

Reviewer 4 Report
As a reviewer I have the following remarks
Line 31. I suggest to write as follows: “(PM10 and PM2.5, coarse and fine, respectively)”. Table 1. Please add a note with definition of: winter, and others. Say Winter: December-February, spring: March-April, or whatever was used. In Australia seasons are different. Table 3 (now is Table 1). Here the question: do hospital admissions are independent of outpatient visits? I mean they don’t overlap – the same persons used twice? Fig 3. If possible, please reduce an empty area, say, shows y-axis in the range 0.95- 1.2, Fig 4. The same idea. In Line 101, please specify used subroutine in SAS. Line 95. Working day vs. holidays, or just week days vs. weekend days? Or additionally holydays?
Thank you.
Author Response
Rewiever 4
Line 31. I suggest to write as follows: “(PM10 and PM2.5, coarse and fine, respectively)”.
Done, kindly see the current version (now line 32-33)
Table 1. Please add a note with definition of: winter, and others. Say Winter: December-February, spring: March-April, or whatever was used. In Australia seasons are different.
Thank you for this comment. It is done, kindly see the current version
Table 3 (now is Table 1). Here the question: do hospital admissions are independent of outpatient visits? I mean they don’t overlap – the same persons used twice?
Yes, it may happen if the person during the outpatient visit was referred to the hospital.
Fig 3. If possible, please reduce an empty area, say, shows y-axis in the range 0.95- 1.2
Done, kindly see the current version.
Fig 4. The same idea.
Done, kindly see the current version.
In Line 101, please specify used subroutine in SAS.
The GENMOD procedure was used with dist=poisson and dscale options.
Line 95. Working day vs. holidays, or just week days vs. weekend days? Or additionally holydays?
Only week days vs. weekend days
Round 2
Reviewer 2 Report
The authors have responded to the comments with appropriate corrections and improvements.